# Interplay of Jahn-Teller Ordering and Spin Crossover in Co(II) Compounds

**Sophia Klokishner \*** and **Serghei M. Ostrovsky** 

Department of Physics of Semiconductor Compounds "Sergiu Rădăuțan", Institute of Applied Physics,
2028 Chișinău, Moldova; sm_ostrovsky@yahoo.com
\* Correspondence: klokishner@yahoo.com

**Abstract:** The spin crossover phenomena in Co(II) compounds are in the focus of the present paper. A microscopic theoretical approach for the description of spin transitions in mononuclear Co(II) compounds is suggested. Within the framework of this approach there are taken into account two types of interionic interactions that may be operative in the problem such as the electron-deformational interaction and the cooperative Jahn-Teller interaction arising from the coupling of the low-spin state of the Co(II) ion with the tetragonal vibrations of the nearest surrounding. The different role of these interactions in the spin transformation is demonstrated and discussed. On the basis of developed approach a qualitative and quantitative explanation of the experimental data on the temperature dependence of the magnetic susceptibility for the $[Co(pyterpy)_2](PF_6)_2$, $[Co(pyterpy)_2](TCNQ)_2 \cdot DMF \cdot MeOH$ and $[Co(pyterpy)_2](TCNQ)_2 \cdot MeCN \cdot MeOH$ compounds is given.

**Keywords:** low-spin state; high-spin state; cooperative electron-deformational interaction; Jahn-Teller ordering; spin crossover; magnetic susceptibility

## 1. Introduction

Since the moment of the discovery of the phenomenon of spin crossover (SCO) the overwhelming majority of research in the field mainly deals with the experimental and theoretical study of iron(II) compounds [1]. Usually in these systems the iron(II) ions are in the nitrogen octahedral surrounding the symmetry of which is close or can be approximated with a good accuracy by the cubic one. The attractive feature of these systems is that the transition occurs between the states $^1A_1$ and $^5T_2$ which significantly differ in the spin and orbital degeneracy that assures different types of the temperature dependence of the high-spin (*hs*) fraction and makes the transition more pronounced. The experimental studies also show that in most cases the spin transition in iron(II) systems is not accompanied by structural reorganization i.e., by the change in crystal symmetry [1]. Such a conclusion also follows from the study performed in the paper [2] in which it has been obtained that the space group in the examined Fe(II) compounds does not change with temperature. Basing on this one can assume that in iron(II) compounds the deformation which arises from the *ls*-*hs* transition (*ls*—low-spin) on the account of the expansion of the electronic shell is mainly a full symmetric one, and the coupling of the spin crossover iron(II) ions with the tetragonal or trigonal deformations plays a secondary role in the spin transition in these compounds. Qualitatively another picture of spin crossover takes place in cobalt(II) compounds in which the difference between the spins of the participating states $^2E$ and $^4T_1$ is much smaller, and for the *ls*- state the Jahn-Teller effect is relevant. In this case the conditions for observation of the spin transitions are more rigid as compared with those for Fe(II) ions since the interaction of the ground $^2E$ state with the Jahn-Teller tetragonal mode leads to additional stabilization of this state that does not facilitate spin crossover. Moreover, the interplay between the electron-deformational and Jahn-Teller cooperative interactions may lead to new interesting peculiarities in the *ls* → *hs*

transformation. Therefore, last years the problem of spin crossover in cobalt(II) compounds has attracted much attention of researchers and became popular. Paying tribute to all researchers working at the problem of spin crossover in cobalt(II) compounds below only some more recent papers dealing with this problem are mentioned. Thus, in paper [3] the investigations of a cobalt(II) clathrochelate complex, which nearly does not change the molecular volume during the spin transformation, demonstrated that even weak intermolecular interactions can cause a pronounced anticooperativity of spin crossover, which results in a more gradual transition in the solid state than in solution. In paper [4] the $[Co(terpy)_2]_3[NbO(C_2O_4)_3]_2 \cdot 3CH_3OH \cdot 4H_2O$ complex was revealed to demonstrate spin crossover behavior. It was shown that namely the $[NbO(C_2O_4)_3]^{3-}$ ligand facilitates this behavior of the complex. The SCO transformation in $[Co(tpy)_2](CF_3SO_3)_2$ and $[Co(tpyphNO)_2](CF_3SO_3)_2$ compounds was examined in [5]. A gradual transition from the *ls*- to the *hs*-state in the range of 150–400 K is characteristic for the tpy-compound. The magnetic susceptibility of the tpyphNO derivative demonstrates a relative abrupt spin transition in the range of 100–250 K together with antiferromagnetic exchange coupling between the cobalt ion and the nitroxide ligand. A hexagonal cobalt(II) metallacycle and its "lipid packaged" derivative, $[Co_6(R\text{-}bisterpy)_6]X_{12}$ (R = $C_{12}$-Glu), have been synthesized and characterized in paper [6]. These compounds incorporating $C_{12}$-Glu lipid anions gave double-layered honeycomb architectures composed of hexagonal stacked tubular structures, which exhibit spin crossover behavior. Recently the first spin crossover compound in which the cobalt(II) is in the mixed $N_4S_2$ coordination environment has been reported [7]. From the magnetic and structural data it follows that the complex manifests a gradual spin transition between 100K and 250K, and the transition temperature $T_{1/2}$ is about 175 K. Among the important publications on the topic of spin transitions in Co(II) compounds the review paper [8] should be also mentioned. The paper represents the results obtained in the examination of cobalt(II) complexes demonstrating spin crossover and, namely, of those in which $[Co(bpy)_3]^{2+}$ and $[Co(terpy)_2]^{2+}$ (where bpy = 2,2-bipyridine, terpy = 2,2:6,2 terpyridine) constitute the main parts. It should be underlined that the study performed in [8] gave the possibility to deepen the understanding of the main features of spin crossover phenomenon in cobalt(II) compounds.

Relatively recently Prof. K.R. Dunbar and her team [9] have reported several new Co(II)-based SCO complexes with the aim of studying the effects of intermolecular π-stacking of the planar terpy ligands and different radicals on the magnetic properties of the resulting materials. Since the Co(II)-ion can exist in a *ls* orbital doublet state or *hs* orbital triplet state the characteristics of the observed spin transitions are different from those in Fe(II) systems. The smaller change in spin associated with the transition ($\Delta S = 1$) and the possible operation of the Jahn-Teller effect in the states involved in the spin conversion lead to special features of the spin crossover scenario in Co(II) systems [10]. It is evident that along with the cooperative interaction facilitating the spin transition the Jahn-Teller effect in the states of the spin crossover Co-ion should be taken into account. As far as we know a model that accounts for both the spin crossover phenomenon and the Jahn-Teller effect in cobalt(II) compounds has not been elaborated. In the present paper we are going to address this problem in order to explain the observed behavior of the compounds $[Co(pyterpy)_2](PF_6)_2$ (**1**), $[Co(pyterpy)_2](TCNQ)_2 \cdot DMF \cdot MeOH$ (**2**) and $[Co(pyterpy)_2](TCNQ)_2 \cdot MeCN \cdot MeOH$ (**3**) reported in [9].

## 2. The Model

The overwhelming majority of systems demonstrating spin crossover belong to the class of molecular crystals. The vibrations of a molecular crystal can be subdivided into two types: the molecular ones and those of the intermolecular type. The role of these vibrations in the spin transition is different, while the molecular vibrations directly coupled to the electronic shells of the spin crossover ions form the energy spectra of these ions, the intermolecular vibrations transmit the local strains that appear during the spin transition from the *ls*-state to the *hs*-one throughout the crystal and are responsible for cooperativity. The idea that this situation can be described by introduction of two types of springs of different rigidity was explored in a series of our previous papers examining spin transitions [11–16]

and will be also applied below in the present work for the description of the spin crossover phenomena in molecular crystals containing Co(II)-ions.

Thus, a crystal containing a Co(II)-ion in the octahedral cubic surrounding as a structural element is examined. It is assumed that the mechanism responsible for the observed spin conversion is the interaction of the Co ions with two spontaneous lattice strains arising on the transition $^2E \rightarrow {}^4T_1$ and, namely, with the fully symmetric ($A_1$) and tetragonal $E$ one. The interaction with the fully symmetric strain is significant for both *hs* and *ls* configurations. As for the $E$ symmetry strain, it is well known [17] that the interaction with this strain is strong for the *ls* $d^7$ electronic configuration with a single *d*-electron in the *e*-orbital since the corresponding deformation leads to large energy stabilization. For the *hs*-state the effect is less noticeable and can be neglected. Additionally, the experimental X-ray data demonstrate that the structural deformation of the compounds under study corresponds to the compression along the 4-th order cubic axis and, therefore, can be described by the *u*-component of the $E$ type deformation. As a consequence, the model below suggested includes the interaction of the *hs*-state of the Co ions only with the spontaneous fully symmetric (denoted below as $\varepsilon_1 = \left(\varepsilon_{xx} + \varepsilon_{yy} + \varepsilon_{zz}\right)/\sqrt{3}$)) lattice strain, while for the *ls*-state the interactions with both totally symmetric $\varepsilon_1$ and $E_u = \left(2\varepsilon_{zz} - \varepsilon_{xx} - \varepsilon_{yy}\right)/\sqrt{6}$ (further on denoted as $\varepsilon_2$) lattice strains are taken into account.

As in [11–16] below a distinction is made between the intra- and intermolecular spaces, and along with the internal molecular $\varepsilon_1$ and $\varepsilon_2$ strains the corresponding external (intermolecular volume) strains $\varepsilon_3$ and $\varepsilon_4$ are introduced into consideration. The part of the crystal Hamiltonian describing the interaction with the mentioned strains looks as follows:

$$
\begin{aligned}
\mathbf{H}_{st} = \tfrac{1}{2}nc_1\Omega_0\varepsilon_1^2 + \tfrac{1}{2}nc_2\Omega_0\varepsilon_2^2 + \tfrac{1}{2}nc_3(\Omega - \Omega_0)\varepsilon_3^2 + \tfrac{1}{2}nc_4(\Omega - \Omega_0)\varepsilon_4^2 \\
+ \varepsilon_1 v_{hs}\sum_k I_{hs}^k + \varepsilon_1 v_{ls}\sum_k I_{ls}^k + \varepsilon_2 v_2\sum_k I_2^k
\end{aligned}
\tag{1}
$$

where $c_i$ are the bulk moduli for the corresponding strains, $\Omega_0$ is the volume occupied by the cobalt(II) ion and its nearest ligand surrounding, $\Omega$ is the unit cell volume per cobalt(II) ion and $k = 1, \dots, n$ enumerates the cobalt ions in the crystal. The first four terms in Equation (1) describe the elastic energy of the deformed crystal, while the last three terms correspond to the interaction of the *d*-electrons of the Co-ions with the $\varepsilon_1$ and $\varepsilon_2$ deformations, $v_{hs}$ and $v_{ls}$ are the constants of interactions of the cobalt ion with the strain $\varepsilon_1$ in the *hs* and *ls* states, respectively, $v_2$ is the constant of interaction of the cobalt ion with the strain $\varepsilon_2$ in the *ls* state. $I_{hs}^k$, $I_{ls}^k$ and $I_2^k$ are the diagonal matrices that have a dimension of the whole basis of the problem under study. The matrix elements of the matrix $I_{hs}^k$ are 1 and 0 for the *hs* and *ls* configurations, respectively. The diagonal matrix $I_{ls}^k$ can be obtained from the $I_{hs}^k$ matrix by replacing all diagonal vanishing matrix elements by 1 and vice versa. The elements of the diagonal matrix $I_2^k$ are 0 for the *hs* configuration, −1 and 1 for the *u* and *v* components of the *ls*-state, respectively.

Introducing new effective coupling parameters $v_1 = (v_{hs} - v_{ls})/2$ and $v_3 = (v_{hs} + v_{ls})/2$, Equation (1) can be rewritten as:

$$
\begin{aligned}
\mathbf{H}_{st} = \tfrac{1}{2}nc_1\Omega_0\varepsilon_1^2 + \tfrac{1}{2}nc_2\Omega_0\varepsilon_2^2 + \tfrac{1}{2}nc_3(\Omega - \Omega_0)\varepsilon_3^2 + \tfrac{1}{2}nc_4(\Omega - \Omega_0)\varepsilon_4^2 \\
+ \varepsilon_1 v_1\sum_k \tau_k + \varepsilon_2 v_2\sum_k I_2^k + \varepsilon_1 v_3 n
\end{aligned}
\tag{2}
$$

where $\tau_k$ is a diagonal matrix with matrix elements equal to $-1$ and 1 for the *ls* and *hs* configurations, respectively. The eigenvalues of the Hamiltonian (2) represent adiabatic potential sheets corresponding to the *hs* and *ls* states of the Co-ions in the crystal. In order to find the equilibrium positions of the nuclei in these states the minimization over all strains is performed. In this procedure the approximate relations $\varepsilon_3 \approx \varepsilon_1 c_1/c_3$ and $\varepsilon_4 \approx \varepsilon_2 c_2/c_4$ are used [12–14]. These relations account for different elasticity of the molecular and intermolecular spaces undergoing full symmetric and tetragonal deformation in cobalt spin crossover crystals and in fact describe a model system in which the mentioned spaces

are presented by connected parallel springs with different elastic moduli $c_1$, $c_3$ and $c_2$, $c_4$, respectively. Finally, one obtains:

$$\mathbf{H}_{st} = -B\sum_k \tau_k - \frac{J_1}{2n}\sum_{k'}\sum_k \tau_k\tau_{k'} - \frac{J_2}{2n}\sum_{k'}\sum_k I_2^k I_2^{k'} \tag{3}$$

where

$$B = A_1 v_1 v_3, \; J_1 = A_1 v_1^2, \; J_2 = A_2 v_2^2 \tag{4}$$

and

$$A_1 = \frac{c_3}{c_1[c_3\Omega_0 + c_1(\Omega - \Omega_0)]}, \; A_2 = \frac{c_4}{c_2[c_4\Omega_0 + c_2(\Omega - \Omega_0)]} \tag{5}$$

The first term in Equation (3) redetermines the crystal field gap between the *ls* and *hs* states. The second and the third terms in Equation (3) represent the infinite range interactions between the cobalt ions which undergo the spin conversion. The obtained intermolecular interactions correspond to the interaction via the field of long-wave acoustic phonons [18].

The nearest ligand surrounding of the Co(II) ion in the compounds under examination is octahedral and consists of 6 nitrogen atoms, its symmetry slightly differs from a cubic one. Since the mean metal ligand distances are of the order of 2Å, the volume of the cube formed by the six ligands and containing the Co ion in the centre is about 64 Å$^3$. As can be seen, for compounds under examination $\Omega >> \Omega_0$. Since the elastic moduli in the spin crossover compounds satisfy the relations $c_1 >> c_3$, $c_2 >> c_4$, Equation (5) can be rewritten as:

$$A_1 = \frac{c_3}{c_1^2\Omega}, \; A_2 = \frac{c_4}{c_2^2\Omega} \tag{6}$$

As a result, the parameters of cooperative interactions $J_1$ and $J_2$ in fact do not depend on $\Omega_0$ for the compounds under examination.

Besides the interaction of the Co ions with two spontaneous lattice strains above mentioned, the model also accounts for the effects of the crystal field acting on the Co(II) ion, the spin-orbital interaction within the *hs*-state, and the Zeeman interaction. The corresponding Hamiltonian looks as follows:

$$\mathbf{H}_0 = -\frac{3}{2}\kappa\lambda\sum_k S^k L^k I_{hs}^k + \frac{\Delta}{2}\sum_k I_2^k + \mu_B\mathbf{H}\sum_k \left(g_0 S^k - \frac{3}{2}\kappa L^k\right)I_{hs}^k + \mu_B\mathbf{H}\sum_k g_0 s^k I_{ls}^k$$
$$+\Delta_{hl}/2\sum_k \tau_k \tag{7}$$

where $\lambda = -180$ cm$^{-1}$ is the spin-orbit coupling parameter, $\kappa$ is the orbital reduction factor and $S = 3/2$ is the spin of the *hs* cobalt ion. In Equation (7) the first term represents the spin-orbital interaction within the $^4T_1$ orbital triplet of the *hs* Co(II) ion written with the use of the so-called TP isomorphism [19]. It is based at the fact that the matrix elements of the orbital angular momentum within $^4T_1$ basis (originating from the $^4F$ term of a free Co(II) ion) are exactly the same as the matrix elements of $-\frac{3}{2}L$ within the $^4P$ basis. Since in the P-basis the orbital angular momentum is $L = 1$, in Equation (7) the fictitious orbital angular momentum $L = 1$ with the factor $-3/2$ is used.

The second term in Equation (7) describes the splitting of the ground $^2E$ term of the *ls*-Co(II) ion caused by the low symmetry crystal field. The splitting of the lowest $^4T_1$ orbital triplet of the *hs*-Co(II) ion by this field is not taken into account due to the reason below explained. The next two terms in Equation (7) describe the Zeeman interaction for the *hs* and *ls* configurations, respectively, with $s = 1/2$ and $\mu_B$ being the spin of the *ls* Co(II) ion and the Bohr magneton. Since in the octahedral surrounding the *hs*-state of the Co(II) ion is orbitally degenerate, the Zeeman interaction contains both the spin and orbital contributions. Finally, the last term in Equation (7) accounts for the energy gap between the centers of gravity of the *hs*- and *ls*-multiplets or in other words the energy gap between the lowest cubic $^4T_1$ term and the ground cubic $^2E$ term. The initial energy gap $\Delta_0$ between the *hs* and *ls* states is redefined with the proper account of the term $-2B$ (see Equation (3)), so in all subsequent

calculations the effective energy gap $\Delta_{hl} = \Delta_0 - 2B$ is used. Thus, the total Hamiltonian of the crystal looks as follows:

$$\mathbf{H} = \mathbf{H}_0 + \mathbf{H}_{st} + \mathbf{H}_{ev} + \mathbf{H}_v \qquad (8)$$

where $\mathbf{H}_{ev}$ and $\mathbf{H}_v$ are the Hamiltonians of the electron-vibrational interaction and free molecular vibrations, respectively. These terms are introduced in Hamiltonian (8) since the cobalt(II) ions in octahedral surrounding interact with the 15 vibrations of this surrounding in both the *ls-* and *hs-* states. At the same time the electron-vibrational coupling does not mix the ground *ls* and excited *hs* states as well as these states with other electronic states. The problem of cooperative interactions arising from the coupling of Co ions with the strains $\varepsilon_1$ and $\varepsilon_2$ is further solved in the mean-field approximation. In this approximation the Hamiltonian (3) is represented by the sum of single-ion Hamiltonians:

$$\mathbf{H}_{st} = -( B + J_1 \, \bar{\tau} \, ) \sum_k \tau_k \, - J_2 \bar{I}_2 \, \sum_k I_2^k \qquad (9)$$

where $\bar{\tau} = Tr(\rho\tau_k)$, $\bar{I}_2 = Tr\left(\rho I_2^k\right)$ play the role of the order parameters and $\rho$ is the density operator:

$$\rho = \sum_k |\varphi_k\rangle \frac{\exp\left(-\frac{E_k}{k_B T}\right)}{Z} \langle\varphi_k| \qquad (10)$$

In Equation (10) the summation runs over all states of the system with $E_k$ being the corresponding energies, $Z$, $k_B$ and $T$ are the partition function, Boltzmann constant and temperature, respectively. From Equations (8) and (9) it follows that the total wave functions of the *ls* and *hs* states can be presented as products of the electronic and vibrational parts, and, hence, the partition functions for these states look as follows:

$$Z_{is} = Z_{is}^{el} Z_{is}^{vib} \; (is \; = \; hs \text{ or } ls) \qquad (11)$$

The vibrational partition functions are:

$$Z_{is}^{vib} = \left(\frac{1}{2\sinh(\hbar\omega_{is}/2k_B T)}\right)^n \; (is \; = \; hs \text{ or } ls), \qquad (12)$$

where $n$ is the number of the normal modes for the Co(II) complex, and the frequencies of all normal modes are replaced by some averaged frequency in the corresponding spin state (*hs* or *ls*). As it was already above mentioned for the complex under study composed of the Co(II) ion and 6 nearest nitrogen donor atoms $n$ is equal to 15. On the basis of density functional theory (DFT) calculations typical values of the averaged frequencies for Co(II) complexes are expected to be about 100 cm$^{-1}$ with the frequency shift between *ls* and *hs* states not more than 15% [8]. In the subsequent calculations we set $\hbar\omega_{hs} = 95$ cm$^{-1}$ and $\hbar\omega_{ls} = 105$ cm$^{-1}$. The difference between these frequencies is about 10%. The latter value does not contradict the information published in paper [8], since in fact in this review only an approximate estimation of the upper limit of this difference is given.

## 3. Estimation of the Characteristic Parameters of the System

For the calculation of the effective coupling parameter $v_l$ of the interaction of the Co-ion with the internal strain $\varepsilon_1$ we use the procedure suggested in [11–16]. The matrix elements $v_{hs}$ and $v_{ls}$ of the operator of interaction with the full symmetric $\varepsilon_1$ strain in the *hs* and *ls* states are:

$$v_{is} = \langle is| \left(\frac{\partial W(r,R)}{\partial R}\right)_{R=R_{is}} |is\rangle \left(\frac{\partial R}{\partial\varepsilon_1}\right)_{R=R_{is}} \; (is \; = \; hs \text{ or } ls), \qquad (13)$$

where $R_{hs}$ and $R_{ls}$ are the metal-ligand distances in the *hs* and *ls* states, and $v_{hs}$ and $v_{ls}$ can be expressed through the mean values of the derivatives of the crystal field energy in these states. For an

octahedral complex $CoX_6$ with the symmetry slightly different from the cubic one, the values $v_{hs}$ and $v_{ls}$ corresponding to the electronic configurations $t_2^6 e$ and $t_2^5 e^2$ are proportional to the cubic crystal field parameters $Dq^{ls}$ and $Dq^{hs}$, respectively, and can be written as $v_{ls} = 90 Dq^{ls} / \sqrt{3}$, $v_{hs} = 40 Dq^{hs} / \sqrt{3}$ (for details see [11], where the corresponding procedure is presented for the spin crossover Fe(II) ions). For crystal field parameters $Dq^{ls} = 1670 \, cm^{-1}$ and $Dq^{hs} = 1300 \, cm^{-1}$ [20] one obtains $v_1 = -2.84 \times 10^4 \, cm^{-1}$. In the compounds under study, the unit cell volumes per Co ion are $\Omega = 1112 \, \text{Å}^3$, $1458 \, \text{Å}^3$ and $1458 \, \text{Å}^3$ for **1**, **2** and **3**, respectively [9]. The typical values of the bulk moduli for cobalt (II) SCO compounds are $c_1 = 7.68 \times 10^{11} \, dyn/cm^2$ and $c_3 = 10^{11} \, dyn/cm^2$ [21]. As a result one obtains that $J_1 = 24.4 \, cm^{-1}$ for **1** and $J_1 = 18.6 \, cm^{-1}$ for **2** and **3**.

Using the results of [22,23] the constant $v_2$ characterizing the coupling with the strain $\varepsilon_2$ in the *ls*-state is calculated with the aid of the relation

$$v_2 = v_{Eu} q_{Eu} / \varepsilon_2 \tag{14}$$

where the operator $v_{Eu}(\mathbf{r})$ possessing the dimension of energy and characterizing the interaction of the Co ion with the $E_u$ vibration of the local surrounding can be written as:

$$v_{Eu}(\mathbf{r}) = \sum_{p,i} \frac{\partial W(\mathbf{r}_i - \mathbf{R}_p)}{\partial q_{Eu}} \bigg|_{q_{Eu}=0} = \sqrt{\frac{\hbar\omega_E}{f_E}} \sum_{p,i} \frac{\partial W(\mathbf{r}_i - \mathbf{R}_p)}{\partial \mathbf{R}_p} \bigg|_{\mathbf{R}_p = \mathbf{R}_p^0} \mathbf{U}_p^{Eu} \tag{15}$$

where $W(\mathbf{r}_i - \mathbf{R}_p)$ is the potential energy of the interaction of the $i$th electron of the Co ion and the $p$th ligand placed at the position $\mathbf{R}_p^0$, $\mathbf{U}_p^{Eu}$ is the unitary matrix for the transformation of the Cartesian displacements $\Delta\mathbf{R}_p$ into the dimensionless coordinate $q_{Eu}$ [22,23], $\omega_E$ is the frequency of the $E$ vibration and $f_E$ is the force constant corresponding to this vibration. Calculating the crystal field potential in the framework of the exchange charge model of the crystal field [24,25], for an octahedral complex $CoX_6$ one obtains the operator $v_{Eu}$ (Equation (15)) in the following form [22]:

$$\begin{aligned} v_{Eu}(\mathbf{r}) = \frac{e^2}{60 R^6} \sqrt{\frac{\hbar\omega_E}{f_E}} \sum_i \sqrt{\frac{\pi}{3}} \Big( &-72\sqrt{5} R^2 Y_{20}(\vartheta_i, \varphi_i) \big[ 3Z\langle r^2 \rangle + 2R^2 G\{S_2(R) - RS_2'(R)\} \big] \\ &+ \big[ \sqrt{70} Y_{4-4}(\vartheta_i, \varphi_i) - 10 Y_{40}(\vartheta_i, \varphi_i) + \sqrt{70} Y_{44}(\vartheta_i, \varphi_i) \big] \big[ 25Z\langle r^4 \rangle + 18 R^4 G\{S_4(R) - RS_4'(R)\} \big] \Big) \end{aligned} \tag{16}$$

where $Ze$ is the effective charge of the nitrogen ligand, $R$ is the distance between the Co-ion and this ligand, $S_l(R)$ and $S_l'(R)$ ($l = 2,4$) are the overlap integrals and their derivatives with respect to the cobalt-ligand distance [24,25]. These integrals are calculated with the aid of double zeta wave functions of cobalt and nitrogen [26]. The values $\langle r^2 \rangle = 1.251$ a.u. and $\langle r^4 \rangle = 3.655$ a.u. for the Co(II) ion are taken from [27]. For the ligand –metal distance $R$ that enters in Equation (16) the mean values $R = 2.042 \, \text{Å}$ for **1** and $R = 2.057 \, \text{Å}$ for **2** and **3** determined from experimental data [9] are accepted. The only phenomenological parameter $G$ was obtained from the cubic crystal field parameter $Dq$ for a transition metal ion in octahedral surrounding [22]

$$Dq = -\frac{2\big(5Ze^2\langle r^4 \rangle + 18 R^4 G S_4(R)\big)}{135 R^5} \tag{17}$$

which represents 1/10 of the difference in the energies of the $e$ and $t_2$ orbitals of the 3d electron for the *ls*-Co-ion. The values of the only phenomenological parameter $G$, that corresponds to $Dq^{ls} = 1670 \, cm^{-1}$ [20], are calculated to be 8.192 for **1** and 8.588 for **2** and **3**.

The vibronic coupling constant $v_{Eu}$ characterizing the interaction of a *ls* Co(II)-ion with the local vibrations of $E_u$ symmetry can be calculated as a matrix element of the $v_{Eu}(\mathbf{r})$ operator (Equation (16)) between the states of the ground orbital doublet of the *ls* Co(II)-ion:

$$
\begin{aligned}
\left|t_2^6 e^2 E_u, M_s = 1/2\right\rangle &= \left|\xi\bar{\xi}\eta\bar{\eta}\varsigma\bar{\varsigma}u\right| \\
\left|t_2^6 e^2 E_v, M_s = 1/2\right\rangle &= \left|\xi\bar{\xi}\eta\bar{\eta}\varsigma\bar{\varsigma}v\right|
\end{aligned}
\tag{18}
$$

The typical value of the force constant $f_E$ is about $10^5$ dyn/cm. As a result, one obtains for the *ls* Co(II)-ion the vibronic parameter $v_{Eu} = 1042$ cm$^{-1}$ for all three compounds. Then with the aid of Equation (14), it can be derived the explicit relation between the vibronic coupling constant $v_{Eu}$ and the parameter $v_2$ characterizing the coupling with the strain $\varepsilon_2$

$$
v_2 = \sqrt{2}R\sqrt{\frac{f_E}{\hbar\omega_E}}v_{Eu}
\tag{19}
$$

The evaluation of the constant $v_2$ of interaction with the strain $\varepsilon_2$ gives the value $6.6 \times 10^4$ cm$^{-1}$. Then, with the parameters $\Omega = 1112$ Å$^3$ (**1**) or $1458$ Å$^3$ (**2** and **3**), $c_2 = 7.68 \times 10^{11}$ dyn/cm$^2$ and $c_4 = 10^{11}$ dyn/cm$^2$ exactly the same as taken above in the calculations of the parameter $J_1$ one obtains that $J_2 = 132$ cm$^{-1}$ for **1** and $J_2 = 100.7$ cm$^{-1}$ for **2** and **3**. The accepted equality of the numerical values of the elastic moduli $c_1 = c_2$ and $c_3 = c_4$ is a reasonable approximation, since for one and the same material the elastic moduli for different type deformations are expected to be values of the same order of magnitude.

## 4. Results and Discussion

The experimental values of the $\chi T$ product for all three complexes are presented in Figure 1 as symbols. As can be seen, even at low temperatures the experimental $\chi T$ values are higher than that expected for the *ls*-Co-ions (for spin s = 1/2 and $g_0 = 2.0$ this product is 0.375 cm$^3$ K mol$^{-1}$). The deviation of the g-factor from the pure electronic for the low-spin Co(II) can be neglected since in the octahedral surrounding the ground state for this configuration is orbital doublet $^2E$ with the matrix elements of the orbital angular momentum within this doublet being zero. The contribution to the g-factor due to the spin-orbital admixture of some other state to the ground $^2E$ one is also negligible because the corresponding energy gaps are large. So, to explain the low temperature values of the $\chi T$ product, it was assumed that in all compounds some number of Co ions do not participate in the spin transition and are from the very beginning in the *hs* state at all temperatures. The fraction of these Co complexes is denoted as $y_{hs}$. The magnetic behavior provided by the Co(II)-ions passing with temperature from the *ls*-state to the *hs*-one is calculated with the use of the model above presented. In further examination for the parameters of cooperative interactions arising from the coupling with the totally symmetric and tetragonal deformations the above estimated values $J_1 = 24.4$ cm$^{-1}$ and $J_2 = 132$ cm$^{-1}$ for **1**, $J_1 = 18.6$ cm$^{-1}$ and $J_2 = 100.7$ cm$^{-1}$ for **2** and **3** were taken. The value of the orbital reduction factor for the *hs*-Co(II)-ion was fixed to its mean value $\kappa = 0.8$. As a result, three parameters and, namely, the effective energy gap $\Delta_{hl}$, the low-symmetry crystal field parameter $\Delta$ and the initial *hs* fraction $y_{hs}$ play the role of fitting parameters. The calculated temperature dependence of $\chi T$ products for all complexes under study are presented in Figure 1 as solid lines. The values of the parameters used in the calculations represent a part of the Figure caption. Comparing the obtained sets of the best fit parameters for all three compounds examined one can notice that these parameters reasonably describe the course of the experimental $\chi T$ curves under study and change reasonably from one compound to another. From Figure 1 it is clearly seen that: (i) the calculated fractions of ions, which are in the *hs*-state from the very beginning, are in good agreement with the observed ones. In fact the inequality between the values $y_{hs}(\mathbf{1}) > y_{hs}(\mathbf{2}) > y_{hs}(\mathbf{3})$ obtained with the aid of the best fit procedure is confirmed by the experimental data; (ii) the relation between the gaps $\Delta_{hl}$ obtained from the fitting

is also reasonable. The gaps obey the inequality $\Delta_{hl}(\mathbf{1}) < \Delta_{hl}(\mathbf{3}) < \Delta_{hl}(\mathbf{2})$ that leads to the situation in which starting from T = 200K the highest is the $\chi T$ curve for compound **1** and the lowest one is the $\chi T$ curve for compound **2**. Thus, this result is also in line with the observed magnetic characteristics; (iii) the obtained negative values of the parameter $\Delta$ corresponds to the axial compression of the local octahedron (stabilization of the $v$ component of the $^2E$ orbital doublet) that agrees well with the experimental observations [9]; (iv) the calculated parameters of electron-deformational interaction are also in line with the experimental data.

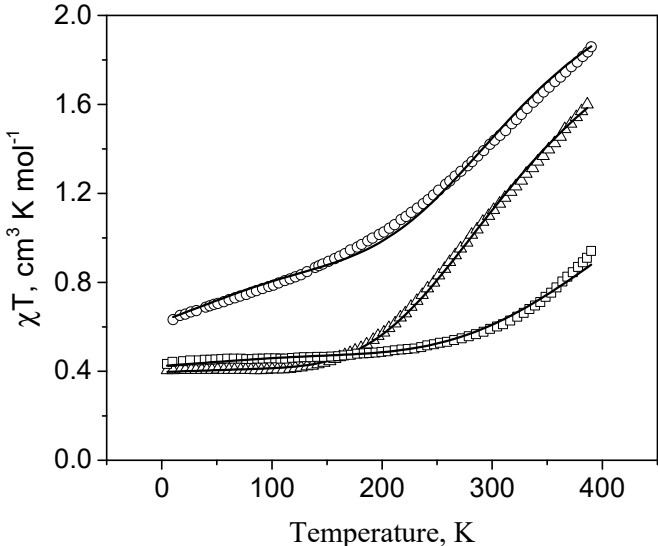

**Figure 1.** Experimental $\chi T$ vs. $T$ dependence for **1** (open circles), **2** (open squares) and **3** (open triangles). The theoretical curves are calculated with $\Delta_{hl} = 885\ \mathrm{cm}^{-1}$, $\Delta = -300\ \mathrm{cm}^{-1}$, $y_{hs} = 20.4\%$ (**1**), $\Delta_{hl} = 1264\ \mathrm{cm}^{-1}$, $\Delta = -300\ \mathrm{cm}^{-1}$, $y_{hs} = 4.0\%$ (**2**) and $\Delta_{hl} = 894\ \mathrm{cm}^{-1}$, $\Delta = -300\ \mathrm{cm}^{-1}$, $y_{hs} = 1.8\%$ (**3**).

In Figure 2 along with the *hs*-fraction calculated as a function of temperature the variation of the order parameters with temperature is presented. It is seen that with temperature increase the parameter $\bar{I}_2$ characterizing the Jahn-Teller distortion falls in magnitude for all compounds. However, at low temperatures up to 150 K its value remains practically constant and close to 1. In the same range of temperatures the mean distortion $\bar{\tau}$ facilitated by the full symmetric deformation acquires the value close to -1. From this it follows that the strong distortion caused by the Jahn-Teller tetragonal mode leads to the stabilization of the *ls*-state, and as a result the population of the *hs*-state is vanishing (with the neglect of the fraction that does not participate in the spin transition). With temperature rise the Jahn-Teller ordering assured by the coupling of Co-ions with the tetragonal mode starts destroying that is expressed in the fall of $\bar{I}_2$, and immediately both the parameter $\bar{\tau}$ and the high spin fraction start to increase. At the same time even at temperatures higher than 350 K the value of the order parameter $\bar{I}_2$ for all studied complexes is not vanishing that indicates that the symmetry is not cubic. All three complexes remain distorted that is confirmed by the structural data [9].

Some comments on the neglect of the effect of the low-symmetry (non-cubic) crystal field for the *hs* configuration should be done. With the aim to compare the splitting within the $t_2$ and $e$ orbitals during the axial compression of the local octahedron, formed by the ligands of the Co(II)-ion, some sample calculations have been performed in the framework of the exchange charge model of the crystal field [24,25]. The performed calculations evidently demonstrated that for reasonable values of the parameter $G$ that characterizes the effects of covalence in the exchange charge model of the crystal field employed in our work the splitting of the $e$-orbital ($\sim300\ \mathrm{cm}^{-1}$) significantly exceeds that ($\sim50\ \mathrm{cm}^{-1}$) of the $t_2$-orbital. Therefore, in the calculations the latter splitting was neglected.

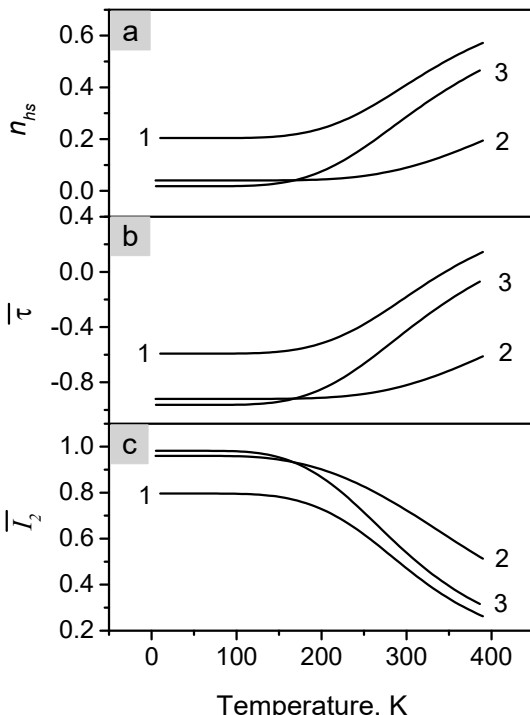

**Figure 2.** Thermal variations of the *hs*-fraction (**a**) and order parameters $\bar{\bar{\tau}}$; (**b**) $\bar{I}_2$; (**c**) calculated with the same set of parameters as in Figure 1.

Finally, resuming the results obtained one can conclude that the picture of spin transformation in Co(II) compounds is different from that in iron(II) ones wherein responsible for the spin transition it is only the interaction with the totally symmetric deformation in the *ls*- and *hs*-states. To describe the observed temperature increase of the magnetic susceptibility in Co(II) compounds along with the interaction with the full symmetric deformation accompanying the spin transition the interaction with the tetragonal mode for the *ls*-state it was necessary to introduce in the developed model. It has been demonstrated that these two interactions play a different role in the spin transformation in Co(II) compounds and compete with one another. The coupling with the full symmetric strain reduces the distance between the states participating in the transition and in fact facilitates the transition. The role of the tetragonal mode is different since it splits the ground *ls E*-level, increases the energy gap between the states participating in the transition and leads in main to gradual type transitions in cobalt (II) compounds.

**Author Contributions:** Writing—original draft, S.K. and S.M.O. All authors have read and agreed to the published version of the manuscript.

**Funding:** This research was funded by the National Agency for Research and Development of Moldova (project 20.80009.5007.19).

**Conflicts of Interest:** The authors declare no conflict of interest.

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
