# Peer review of "Interplay of Jahn-Teller Ordering and Spin Crossover in Co(II) Compounds"

_magnetochemistry, doi:10.3390/magnetochemistry6040062_

Round 1

Reviewer 1 Report

The scientific content of the ms. is referring to a theoretical approach for the description of spin transitions in mononuclear CoII compounds. The present study is well-written and a detailed theoretical analysis has been performed.  However, to my opinion minor revision of this ms. is required concerning the content and thus some points should be examined/ reconsidered by the authors in order to this article be accepted.

General comments on this experimental work – which supports my proposal for acceptance after minor revision – and revision points/comments/suggestions to be taken into account:

(a) Introduction: The authors provide with detail recent efforts on the topic of spin crossover cobalt(II) complexes at the inter/intramolecular interactions point of view. At the last paragraph, lines 67-71, the authors refer to a recent study focused on Co(II)-based complexes in which supramolecular interactions are present (“Relatively recently Prof. K.R. Dunbar and her team [8] have reported several new Co(II)-based SCO complexes with the aim of studying the effects of intermolecular π-stacking of the planar terpy ligands and different radicals on the magnetic properties of the resulting materials. However, paper [8] contains only experimental data, and the explanation of the observed transformations in the magnetic and structural data is absent”). This last statement is too strong; the research activity is presented as a deficient effort which do not cover its scope and thus authors should rephrase it.

(b) Results and discussion: The authors describe in detail the followed methodology which is very important to unfamiliar readers. Towards the aim of this work they have chosen the SCO complex [Co(pyterpy)2](PF6)2. However, the authors to not mention why they have chosen this complex to develop the theoretical based model. The authors could add a relevant explanation in the ms.

In addition, it is important to clarify (if possible) if this complex could be considered as a representative example and an analogous analysis could be also performed for other SCO compounds subjected simultaneously to Jahn-Teller effect and spin crossover phenomenon. If this is feasible it should be added in the ms to enhance the importance of the study.

Reviewer 2 Report

Authors of the article entitled "Interplay of Jahn-Teller ordering and spin crossover in Co(II) compounds" presented a promising theoretical model for the description of spin crossover phenomena in cobalt(II) compounds. They also showed practical application of this model for [Co(pyterpy)2](PF6)2 complex. Moreover, all analyses have been supported by literature references.

At first glance, the presented results seem consistent, however, this changes after an in-depth look at the model and analysis. The major drawback of this study is insufficient explanation how to determine values of used parameters (e.g., Ω0). Authors report that: Ω0 for [Co(pyterpy)2](PF6)2 compound is approximately 64 Å3 without explanation of its origin and the error margin. Moreover, the authors do not provide error margins for other parameters which makes it impossible to determine the precision of this model. Next, this model has been tested only for one specific system showing incomplete gradual spin crossover and for the input parameters determined at certain temperature, undescribed in the text. The authors should test the model on other systems with more abrupt and clear transition (examples can be found in the uncited review article: Coord. Chem. Rev. 2011, 255, 1981-1990) or for other two compounds reported in the same article as [Co(pyterpy)2](PF6)2 (J. Mater. Chem. C, 2015,3, 9292-9298). Furthermore, they should also perform additional calculation using crystal structures at different temperatures to prove that different starting values of the unit cell volume per Co ion (Ω) do not affect on results, if so they should specify at which conditions the Ω value should be determined. Finally, the authors do not refer to the value g-factor which leads to the assumption that the expected χT value at low spin state adopts 0.375 cm3Kmol-1 (g = 2.0), and it implicates next assumption that remaining 0.235 cm3Kmol-1 magnetic signal at 5 K can be addressed to “some number of Co ions do not participate in the spin transition and are from the very beginning in the hs state at all temperatures”. Author should consider larger values of g-factor (g = 2.55 results in χT = 0.61 cm3Kmol-1), since this factor can reach a maximum value of 13/3 (Inorg. Chim. Acta 2008, 361, 3432-3445).

Concluding, this paper in recent form is not suitable for publication in Magnetochemistry and it demands major revision.

Reviewer 3 Report

The paper is reasonably well written. Although the details of the model cannot be understood without consulting the previous work of the authors, the manuscript contains sufficient information to catch the main points. There are a few points that should be addressed to improve the quality.

1. I dearly miss an exploration of an analysis of how sensitive are the results to small changes in the parameters. Being a semi-empirical model with quite some (mostly very sensible) approximations, it might be good to what extent fitting the magnetic susceptibility can decide on the appropriateness of the model. There may be many sets of very different parameters that equally well fit the curve. If this is the case, the validity of the model would be doubtful.

2. The relation between G and 10 Dq is not clear. There is no indication how G was determined from 10Dq. Moreover, the text is repetitive; same message in line 213 and 219

3. The term "fictitious orbital moment" and the choice (?) for L=1 must be explained. Is it because of a partial orbital momentum quenching of the 4T? This would be in contrast with neglecting the t-orbital splitting in the hs state.

4. The difference of the t and e splitting is not very large. Actually, the value of the t splitting reported in the manuscript is surprisingly large. Please comment.

5. Change the wording in line 227: "is a sufficiently rough approximation" seems odd. I guess the opposite is meant.

6. The introduction states that in Fe(II) complexes only the symmetric deformation is relevant. The authors may want to check the articles written by Alvarez et al. on this topic (for example: J. Am. Chem. Soc. 125, 6795 and Chem. Eur. J. 16, 10 380)

Round 2

Reviewer 2 Report

The authors of the manuscript entitled: "Interplay of Jahn-Teller ordering and spin crossover in Co(II) compounds" responded to the comments, and they modified manuscript. The answers were adequate, thus, this paper is suitable for publication in Magnetochemistry in its latest form.

Reviewer 3 Report

The answers are to the points raised by the reviewers are sufficient. The validity (robustness) of the fitted parameters has become more clear. No further comments